# Stem Cell-Secreted Allogeneic Elastin-Rich Matrix with Subsequent Decellularization for the Treatment of Critical Valve Diseases in the Young

**DOI:** 10.3390/bioengineering9100587

**Published:** 2022-10-20

**Authors:** Brittany A. Gonzalez, Ariadna Herrera, Claudia Ponce, Marcos Gonzalez Perez, Chia-Pei Denise Hsu, Asad Mirza, Manuel Perez, Sharan Ramaswamy

**Affiliations:** Department of Biomedical Engineering, Florida International University, Miami, FL 33174, USA

**Keywords:** heart valves, oscillation shear stress, elastin, decellularization, accelerate regeneration

## Abstract

Critical valve diseases in infants have a very poor prognosis for survival. Particularly challenging is for the valve replacement to support somatic growth. From a valve regenerative standpoint, bio-scaffolds have been extensively investigated recently. While bio-scaffold valves facilitate acute valve functionality, their xenogeneic properties eventually induce a hostile immune response. Our goal was to investigate if a bio-scaffold valve could be deposited with tissues derived from allogeneic stem cells, with a specific dynamic culture protocol to enhance the extracellular matrix (ECM) constituents, with subsequent stem cell removal. Porcine small intestinal submucosa (PSIS) tubular-shaped bio-scaffold valves were seeded with human bone marrow-derived mesenchymal stem cells (hBMMSCs), cultured statically for 8 days, and then exposed to oscillatory fluid-induced shear stresses for two weeks. The valves were then safely decellularized to remove the hBMMSCs while retaining their secreted ECM. This de novo ECM was found to include significantly higher (*p* < 0.05) levels of elastin compared to the ECM produced by the hBMMSCs under standard rotisserie culture. The elastin-rich valves consisted of ~8% elastin compared to the ~10% elastin composition of native heart valves. Allogeneic elastin promotes chemotaxis thereby accelerating regeneration and can support somatic growth by rapidly integrating with the host following implantation. As a proof-of-concept of accelerated regeneration, we found that valve interstitial cells (VICs) secreted significantly more (*p* < 0.05) collagen on the elastin-rich matrix compared to the raw PSIS bio-scaffold.

## 1. Introduction

Congenital heart defects occur in about 1% of live births and is a relevantly prevalent birth anomaly [1]. A sub-set of infants are born with a critical congenital valve disease (CCVD), which currently has no effective treatment measure other than a heart transplant, which is rarely available, or possibly via a compassionate care option based on an experimental treatment. The primary reason for the lack of treatment options is due to the lack of available prosthetic valves in small sizes and their inability to support somatic growth. A valve with regenerative capacities would thus be especially appealing since these barriers could be overcome. Specifically, such a valve could facilitate growth, self-repair, infection resistance and potentially be a permanent approach for resolving CCVD in children.

Multiple efforts to facilitate regenerable valves have focused on decellularized, extracellular matrix (ECM) bio-scaffolds [2,3,4,5,6,7]. One example of a decellularized bio-scaffold material that has received considerable attention for use in cardiovascular applications, and which has also received Food and Drug Administration (FDA) Investigational Device Exemption (IDE) for early clinical feasibility assessment for tricuspid valve replacement in adults, is porcine small intestinal submucosa (PSIS) [2,8,9,10].

The major concern with PSIS has been an adverse immune response [6]. We also recently found that following valve replacement, a 3-month window exists for PSIS in which complete valve regeneration has to occur on the bio-scaffold and if not, a hostile immune response would occur, leading to valve failure [11].

Previously, we showed that oscillatory flow culture of human bone marrow mesenchymal stem cells (hBMMSCs) supported the valve phenotype [12,13,14]. For the more effective treatment of CCVD in children, the goal of this study was to (i) seed hBMMSCs onto cylindrical PSIS valves to deposit a thin layer of allogeneic engineered ECM on the bio-scaffold during oscillatory flow culture (ii) safely remove the hBMMSCs from the bio-scaffold valves, i.e., decellularization and (iii) characterize the engineered ECM to identify components that would serve as a trigger to accelerate regeneration, in order to facilitate complete valve regeneration following implantation within a 3-month window.

Of relevance, several studies have described the importance of elastin within the ECM material to trigger in vivo chemotaxis, leading to native cardiovascular cell migration to the site of the implant [15,16,17,18,19,20]. However, there are still some limitations associated with effective bio-scaffold ECM-to-native cell communications if the heart valve scaffold contains xenogeneic elastin [21]. While allogeneic scaffolds have shown promising regenerative findings, including for the heart valve application, there is the obvious limitation of being able to harvest bio-scaffolds from human donors. However, if we can effectively deposit engineered ECM with rich-allogeneic, i.e., human elastin, content onto a xenogeneic bio-scaffold using hBMMSCs, which are widely commercially available, this approach would alleviate the concerns of using solely xenogeneic bio-scaffolds for valve replacement. In addition, there are legitimate concerns with leaving living biologics in place, such as the stem cells, which leads to an over-abundance of tissue formation and thereby, causing adverse events such as leaflet fusion with the surrounding vasculature [22]. This concern can be eliminated if the stem cells can be safely removed from the valve construct prior to implantation, while retaining the stem cell secreted engineered human ECM. This effective stem cell decellularization following the stem secretion of effective allogeneic ECM on the valve bio-scaffold was therefore another major objective of the present study.

## 2. Materials and Methods

A custom-built, in-house “torpedo” bioreactor (Figure 1A,B) [23] was utilized to apply oscillatory flow of culture media onto tubular PSIS bio-scaffold valves (CorTricuspid ECM valve, CorMatrix, Roswell, GA, USA). The flow profile used was a square waveform with 17% positive (forward) flow and 83% negative (backwards) flow direction (Figure 1C–E), which resulted in a time-averaged wall shear stress (TAWSS) of 3.1 dynes/cm^2^. The magnitude of the flow oscillations was quantified via the oscillatory shear index (OSI) metric, previously defined by He and Ku [24] as being between 0 ≤ OSI ≤ 0.5 with 0 being unidirectional (steady) flow and 0.5 being flow with maximum oscillations. Here, the OSI was 0.2, which we have previously established to be of physiologic-relevance to heart valves [25].

### 2.1. In Vitro Cellular Culture of Human Bone Marrow Stem Cells

hBMMSCs (RoosterBio, Inc., Frederick, MD, USA; catalog #: MSC-003) were culture expanded following the manufacturing protocol. In brief, frozen hBMMSCs were thawed and transferred to a 50mL centrifuge tube, with the gradual addition of 4 mL of prepared culture media (h-MSC High Performance Basal Medium (RoosterBio, Inc. catalog #: SU-005) and Media Booster GTX (RoosterBio, Inc., catalog #: SU-005) with 1% penicillin and streptomycin (P/S; Cytiva, HyClone™, Fisher Scientific, Pittsburgh, PA, USA; Catalog #: SV300010). The cells were then centrifuged at 200× *g* for 10 min and the supernatant was subsequently carefully removed. The cells were then resuspended in 45 mL of freshly prepared culture media and plated onto a T225 flask (Fisher Scientific, Catalog #: 10-126-63). Once the flask was confluent, the hBMMSCs were harvested with 0.25% Trypsin-EDTA (1X) (Fisher Scientific, Catalog #: 25200-056) and continuously passaged until 28 × 10^6^ cells (at passage 4) was obtained.

### 2.2. Seeding of hBMMSCs on Tubular PSIS Bio-Scaffold Valves

The tubular PSIS bio-scaffold valves (*n* = 4) were placed on a torpedo holder (*n* = 2 holders; *n* = 2 bio-scaffold valves per holder) (see Appendix A) to secure the bio-scaffolds when placed in a conical tube. The bio-scaffolds were rinsed with phosphate-buffered saline (PBS; Gibco™, Fisher Scientific, Catalog #: 10-010-031) for 5 min and transferred to 50 mL vented conical tubes (Product # TP87050, TPP, TubeSpin Bioreactor, Zollstrasse 7, CH-8219 Trasadingen, Switzerland). The harvested hBMMSCs suspension was then added to the conical tubes that contained the PSIS tubular valves (7 × 10^6^ hBMMSCs per bio-scaffold valve) to which, 45 mL of tissue culture media consisting of Dulbecco’s modified Eagle’s medium (DMEM; Fisher Scientific, Catalog #: MT10013CV) supplemented with: 10% fetal bovine serum (FBS; Fisher Scientific, Catalog #: MT35010CV), 1% P/S (Cytiva, HyClone™, Fisher Scientific, Catalog #: SV300010), 2 ng/mL basic fibroblast growth factor (bFGF, Corning™; Fisher Scientific, Catalog #: CB-40060) and 82 μg/mL ascorbic acid 2 phosphate (AA2P; Fisher Scientific, Catalog #: NC0602549) (Appendix A). These tubes were then placed on a rotisserie culture system (LabquakeTM Rotisserie Hybridization Rotators, ThermoFisher Scientific, Waltham, MA, USA) at 10 RPM inside a sterile incubator with 95% air, 5% CO_2_ and controlled humidity for 8 days.

After 8 days in the rotisserie, some of the tubular PSIS valves *(n* = 2, “static” control group) were retained on the torpedo holder in rotisserie culture for an additional 14 days without media change. The other tubular bio-scaffold valves (*n* = 2, dynamic group) were retained on the torpedo holder but were transferred into our torpedo bioreactor (Appendix A) for 14 days, which is similar to a previously used bioreactor for the purpose of conducting heart valve tissue engineering experiments [13,14,25].

### 2.3. Histology of Engineered Tissues Deposited on Tubular Bio-Scaffold Valves

After 22 days (22 days solely rotisserie “static” culture or 8 days rotisserie “static” seeding + 14 days of oscillatory flow culture), the hBMMSC-seeded bio-scaffold valves (*n* = 2 static; *n* = 2 dynamic) were removed (from rotisserie or torpedo bioreactor) and immediately fixed in 10% (*w*/*v*) formalin for 24 h. The valves were then rinsed with PBS and embedded in a slow freeze process with optimal cutting temperature (OCT; OCT, Fisher Scientific, Catalog #: 23-730-571). The embedded valves were then stored at −80 °C overnight. The following day, the embedded samples were cut to a thickness of 16 µm (each slice) and then placed on glass slides (TruBond 380; Newcomer Supply, Middleton, WI, USA), which were set to dry at room temperature overnight.

Hematoxylin and Eosin (H&E; Epredia™ Shandon™ Rapid-Chrome H & E Frozen Section Staining Kit, Fisher Scientific) staining was used to determine the morphology and presence of cells of the valves. A Movat’s stain (Russel-Movat Pentachrome; American MasterTech Scientific, Lodi, CA, USA; catalog #: KTRMP) was then applied to determine the different constituents of leaflet hBMMSC-derived engineered tissues, specifically collagen (yellow/orange), mucins (blue/green) and elastin (purple/black).

### 2.4. Spatial Intensity Mapping of Tubular Valves for Assessment of Percent Extracellular Matrix Content

Spatial intensity mapping of tubular valves was computed as previously reported [11,23]. In brief, a Movat’s image of a strip of the raw, acellular PSIS bio-scaffold (Cormatrix) was used as a reference image for all the image analysis acquired from Movat’s. All the images were then normalized to the thresholded reference image through an in-house script (MATLAB, MathWorks, Inc., Natick, MA, USA). After normalizing, a color segmentation plugin was used (ImageJ, NIH Image, Bethesda, MD, USA) and different colors of yellowish-green, purple-black, bluish-green, and fuchsia were chosen to distinguish the different ECM components of collagen, elastin, proteoglycans, and fibrin, respectively. Once complete, the option of independent color channels was selected, and a Hidden Markov Model was used to quantify the resulting percentages of the ECM concentrations. The images were then processed for their respective signal intensities within each ECM component of interest (MATLAB, MathWorks, Inc.) and spatial intensity maps for each of these ECM components were subsequently generated.

### 2.5. Elastin Immunofluorescence Imaging of hBMMSC-Seeded Constructs after Oscillatory Flow Dynamic Culture

Three specimen groups, i.e., the oscillatory flow conditioned, hBMMSC-seeded PSIS bio-scaffold valves along with static controls (22 days of rotisserie culture) and raw PSIS samples were fixed in 10% formalin at 4 °C overnight, embedded in OCT compound, and sectioned at 16 µm using a cryostat. Sectioned samples were subsequently stained with elastin mouse monoclonal primary antibody (Novus Biologicals, Littleton, CO, Catalog #: NB017430), followed by goat anti-mouse polyclonal secondary antibody (Thermo Fisher, Waltham, MA, USA, Catalog #: PISA510264) and 4′,6-diamidino-2-phenylindole (DAPI). Stained samples from each of the three groups were subsequently imaged using a confocal microscope (Nikon Eclipse Ti, Minato, Tokyo, Japan) and the images were then quantified using signal intensity mapping with a custom in-house written code (MATLAB, MathWorks, Natick, MA, USA).

### 2.6. Decellularization of Seeded-hBMMSCs from the Bio-Scaffold after 22 Days of Tissue Culture

After the net 22-day tissue culture period (22-day rotisserie or 8-day rotisserie + 14-day oscillatory flow conditioning), the hBMMSC-seeded bio-scaffolds were sectioned into strips and immersed in 1% Triton X-100 (Fisher Scientific, Catalog #: BP151-100) deionized water solution and agitated on a shaker set at 20 RPM for 24 h at room temperature. The constructs were then washed with PBS for 24 h at room temperature, with a fresh PBS change at the 12 h mark.

### 2.7. Histological Assessment and Spatial Intensity Mapping of Decellularized Constructs for Assessment of Cell Removal and Percent Extracellular Matrix Content

hBMMSCs (RoosterBio, Inc.) were cultured until confluent (at passage 4) in cell culture media as described in Section 2.1. The hBMMSCs were harvested and used to seed PSIS (Cormatrix) bio-scaffolds (*n* = 4 samples; 35 × 10^6^ cells per scaffold) and were rotisserie cultured for 8 days in media (RoosterBasal™-MSC, RoosterBio) with 2% supplement (RoosterBooster™-MSC-XF, RoosterBio), 10% exosome-depleted FBS (Gibco™, Fisher Scientific), 2% P/S (Thermo Fisher Scientific), 2 ng/mL bFGF (Corning™; Fisher Scientific Catalog #: CB-40060) and 82 μg/mL AA2P (Sigma-Aldrich, Dorset, UK). On day 6 of rotisserie culture, while EV Boost™ (RoosterBio) was added the culture media, it was done so for an experiment not connected to this study.

After 8 days of rotisserie culture, half the scaffolds were fixed in 10% formalin (*w*/*v*) for 24 h. The other half were decellularized and then fixed in 10% formalin (*w*/*v*). All scaffolds were embedded in OCT and then stored in −80 °C. The embedded samples were cut to a thickness of 16 µm, placed on glass slides (TruBond 380; Newcomer Supply, Middleton, WI, USA), and dried at room temperature overnight. Slides were then subjected to H&E staining (Epredia™ Shandon™ Rapid-Chrome H & E Frozen Section Staining Kit, Fisher Scientific) and a Movat’s stain (Russel-Movat Pentachrome; American, catalog #: KTRMP, MasterTech Scientific) and imaged using bright-field microscopy (Zeiss, Axiovert 40 CFL, Maple Grove, MN, USA). Spatial intensity maps for collagen and elastin were generated from the Movat’s stain images using the same protocol detailed in Section 2.4.

### 2.8. Uniaxial Tensile Testing of Decellularized Constructs

As described in Section 2.7, hBMMSCs were identically cultured and seeded onto PSIS bio-scaffold strips (18 mm × 10 mm × 1 mm; *n* = 12; 5 × 10^6^ cells per scaffold) but with the exclusion of the EV Boost reagent (RoosterBio). After 8 days of rotisserie culture, half of the scaffolds were decellularized (*n* = 6; “TX100” group). The other half (*n* = 6; control group) directly underwent uniaxial tensile testing using our mechanical testing equipment (Electroforce, Bose, San Diego, CA, USA). The specimens were tested until either the system’s maximum displacement or failure was reached during specimen displacement along the following two directions: specimen length and specimen width (*n* = 3/orientation) (Appendix A). The decellularized specimens subsequently underwent the same mechanical testing protocol.

### 2.9. Response of Valve Interstitial Cells to Bio-Scaffolds with and without Engineered ECM

Rat valve interstitial cells (VICs) were cultured in media (DMEM; Corning, Fisher Scientific, Catalog #: MT10013CV), enriched with 10% FBS (Fisher Scientific, Catalog #: MT35010CV) and with 1% P/S (Cytiva, HyClone™, Fisher Scientific, Catalog #: SV300010). The culturing environment was maintained in a sterile incubator environment at 37 °C, 5% CO_2_ until the cells were fully confluent. The collected VICs were subsequently seeded onto the elastin-rich engineered ECM and raw PSIS bio-scaffold substrates (*n* = 4 specimens/group), with a density of 8 × 10^5^ cells/cm^2^ per specimen. The VIC-seeded substrates were then rotisserie cultured for a total of 8 days. The samples were then transferred into the torpedo bioreactor for 7 days of oscillatory flow dynamic culture [12,13,14,25], operating at a TAWSS of 3.1 dynes/cm^2^ and an OSI of 0.2 and within standard cell culture incubator conditions. The media used for the static and dynamic conditioning included DMEM, 10% FBS, 2% P/S, 2 ng/mL bFGF (Corning™; Fisher Scientific, Catalog #: CB-40060) and 82 μg/mL AA2P (Fisher Scientific, Pittsburgh, PA, USA; Catalog #: NC0602549). 

After dynamic culture of the rat VICs on the two substrates, a Masson’s Trichrome Stain (American MasterTech Stat Lab, Lodi, CA, USA, Catalog #: KTMTR) was used to histologically assess the levels of collagen secretion by the VICs on the outer surfaces of the specimens (staining for collagen in blue, cytoplasm and fibers in red, and nuclei in black). In addition, samples were imaged at 40× magnification and collagen was quantified using a custom in-house written signal intensity mapping code (MATLAB, Mathworks). This signal intensity was reported as an average ± standard deviation (SD).

### 2.10. Statistical Methods

Once the spatial intensity maps of the tubular valves were created on MATLAB (MATLAB, Mathworks) from the Movat’s images, the average percent (%) ± the standard error of the mean (SEM) spatial intensity maps was computed (Excel, Microsoft, Redmond, WA) for certain depths as well as across the whole sample. A *t*-test was then performed for the Movat’s staining, signal intensity quantification of engineered ECM components (mucins, collagen, elastin, and fibrin) between the static and dynamic culture groups. Additionally, the percent (%) of unfilled PSIS was calculated by first determining the total area of each image (ImageJ, NIH Image). Then, the regions/image, which exhibited a dark pink stain, i.e., presence of engineered tissues, in each H&E image was computed (Image J, NIH Image). Finally, the remaining area/image, i.e., without any staining, was divided by the total area/image to evaluate the average percent (%) of unfilled tissue ± SEM (Excel, Microsoft). Spatial intensity maps (MATLAB, Mathworks) were also used to quantify the immunostaining for elastin. The spatial intensity values were subsequently compared between groups (raw PSIS, static and dynamic groups), using a one-way ANOVA analysis followed by Tukey’s post hoc assessment (SPSS, IBM, Armonk, NY, USA).

From the uniaxial tensile testing results of the decellularized constructs, the average linear stiffness, average maximum stress, and corresponding SEM of each treatment group in each orientation was calculated (Excel, Microsoft). A *t*-test was then performed to assess if there was a significant difference between the two groups (untreated versus decellularized) investigated. Similarly, a *t*-test was also conducted for the VICs signal intensity maps, to analyze the statistical difference between the raw PSIS bio-scaffold and the elastin-rich ECM groups. 

For all the statistical analyses that were performed in this study, a statistically significant difference between groups was interpreted to have occurred when the *p*-value was <0.05.

## 3. Results

### 3.1. Morphology of Engineered ECM Grown on Tubular Bio-Scaffold Valves

The morphology and cellularity of the static, i.e., 22-day rotisserie culture, and 8-day static + 2 weeks dynamically cultured bio-scaffold valves were assessed via H&E staining. It was found that both the static and dynamic groups had the presence of *de novo* human tissues on the bio-scaffolds and cellular infiltration was observed (Appendix A). It appeared that for both groups, the layers of the bio-scaffold valves towards the outer surface had less tissue and was disorganized (Appendix A), although the dynamic group was more aligned. With increasing tissue-depth, visually, the amount of tissue and cells present increased (Appendix A).

### 3.2. De Novo Extracellular Matrix and Spatial Intensity Quantification of Tubular Bio-Scaffold Valves with Movat’s Pentachrome Staining

The engineered ECM was additionally assessed for both static and dynamic groups with Movat’s Pentachrome histological staining to identify collagen, elastin, fibrin, mucins, and nuclei in yellow, black, intense red, blue to green, and black, respectively. Engineered tissue was found present in both groups, with a denser ECM as the depth increased (16 µm, 304 µm, 592 µm and 880 µm) towards the inner surface of the bio-scaffold valves (Figure 2). The static group was mostly composed of mucins at the lower depths (outer surface of bio-scaffold valves) (Figure 2) with a combination of collagen, fibrin, and elastin. As the depth increased, towards the inner surface of the bio-scaffold valve, there was a predominant presence of collagen (Figure 2). The dynamic group had less tissue components at the outer surface of the bio-scaffold valve (Figure 2) consisting of mucins and collagen. However, as the depth increased towards the inner surface of the bio-scaffold valve (304 µm, 592 µm and 880 µm), the ECM became organized and aligned towards the direction of flow (Figure 2). The presence of collagen, elastin and fibrin was also more abundant at these depths (Figure 2).

Across the whole samples for both static and dynamically cultured hBMMSC-seeded bio-scaffold valves, the average ECM content percentages were computed. The statically cultured valves had an overall ECM content percentage of 18% ± 1.09 mucins, 19% ± 1.40 collagen, 4% ± 0.43 elastin and 7% ± 1.12 fibrin. The dynamically cultured valves had an overall ECM content percentage of 21% ± 0.87, 19% ± 0.98, 8% ± 0.72 and 12% ± 0.85 of mucins, collagen, elastin and fibrin, respectively. 

It was found that the ECM percentage content was significantly different (*p* < 0.05) for all ECM components assessed except collagen (no significant difference; *p* > 0.05). The dynamically cultured valves had a significantly higher (*p* < 0.05) ECM content percentage of mucins, elastin and fibrin compared to the statically cultured valves. Furthermore, the average percent of “unfilled” de novo ECM ± SEM was computed across the whole samples (Appendix A). The static group had 53% ± 2.84 unfilled de novo tissue formation, while the dynamic group had 39% ± 1.83 (Appendix A); a significant difference (*p* < 0.05) was found between the groups, where the dynamic group had a significantly higher (*p* < 0.05) amount of de novo tissue formation in vitro. Finally, to confirm the presence of mature allogenic elastin derived from human stem cell oscillatory flow conditioning, immunofluorescent staining specific to elastin was performed and confirmed a higher presence of elastin (*p* < 0.05) after bioreactor culture compared to statically cultured and raw bio-scaffold specimens (Figure 3).

### 3.3. Morphological Assessment and Spatial Intensity Quantification of Decellularized Scaffolds after Oscillatory Flow Conditioning

The ECM of the decellularized scaffolds was assessed for the presence of cells and compared to the ECM of the untreated scaffolds after oscillatory flow mechanical conditioning. Visually, there was a large presence of cells on the control scaffolds (Figure 4a; purple spots). The decellularized group (Figure 4b), however, appeared to have little to no cells present, indicating that the decellularization protocol was efficient in cell removal. 

The ECM of both the untreated and decellularized groups was quantified with regard to collagen and elastin (Figure 4) content. The average percent of protein content ± SEM was calculated based on all the image slices (Table 1). It was found the decellularized group had a slightly higher elastin amount, with the average elastin of 19.60% ± 3.39 and 22.06% ± 1.95 for the control and the decellularized groups, respectively. There was also an increased amount of collagen content in the decellularized group, with an average of 31.29% ± 2.73 while the control group had 26.40% ± 2.09. There was no significant difference (*p* > 0.05) between the two scaffold groups for elastin nor collagen content.

### 3.4. Uniaxial Tensile Testing Assessment of Decellularized Scaffolds

The mechanical properties of the untreated and the decellularized samples were assessed via uniaxial tensile tests along the specimen width and length directions (Appendix A). The linear stiffness (MPa) and maximum tensile stress (MPa) were calculated for each sample and an average ± SEM (Appendix A) for each property was obtained for each group in each direction. It was found that there was no significant difference (*p* > 0.05) between width-dimension’s linear stiffness, length-dimension’s linear stiffness, width-dimension’s maximum tensile stress, and the length-dimension’s maximum stress (Appendix A).

### 3.5. Collagen Secretion after Valve Interstitial Cells (VICs)-Seeding onto the Engineered Elastin-Rich ECM Compared to VIC-Seeding onto Raw PSIS

A comparison of collagen secretion by VICs after seeding onto the engineered elastin-rich ECM as well as the raw PSIS bio-scaffold was assessed via Masson’s Trichrome Staining on their respective outer surfaces (*n* = 4 samples/group). Results showed an average positive collagen signal intensity of 0.9670 ± 0.070 AU (Arbitrary Units) on the elastin-rich ECM in contrast to raw PSIS that exhibited a VIC-based collagen secretion signal intensity of 0.7481 ± 0.035 AU (Figure 5a). A statistically significant increase (*p* < 0.05) was found in VIC-based collagen secretion when seeded on the elastin-rich ECM (Figure 5b) compared to the equivalent VIC-seeding onto the raw PSIS bio-scaffold (Figure 5c).

## 4. Discussion

CCVD in the young has no viable treatment options currently available mainly due to limitations in current artificial valve sizing options and their inability to support somatic growth. Emerging research in regenerative medicine could potentially address current limitations to treat CCVD in pediatric patients, providing growth, self-repair, infection resistance and thus, an effective, permanent solution to CCVDs in the young [26]. The key to un-tapping the potential in making regenerated valves a reality largely depends on the choice of cell(s), the scaffold material(s), the specific construct preparation protocol (e.g., use of a bioreactor) and the subsequent responses between the implanted construct and its surrounding, living environment. Valve leaflets experience complex, time-varying external loading patterns resulting from local hemodynamic forces. Thus, it is critical that the degradable scaffold possesses suitable material properties and is processed and assembled for optimum, long-term functionality. In this regard, bio-scaffolds such as PSIS do support short-term function in the order of months and are also being used with positive outcomes in compassionate care treatment of infants born with a CCVD, as well a current on-going clinical trial of PSIS tri-cuspid valve repair in adults suffering from tri-cuspid valve leakage [27]. However, to support somatic growth in a young recipient, we as well as others have shown in animal models that chronic inflammation leads to PSIS bio-scaffold valve failure, in the order of a few months after implantation [11,28,29]. This hostile response is a result of small, portions of the bio-scaffold that remain unfilled [11,28,29] with host’s de novo tissues. 

Alternatively, cell culture-based tissue engineering heart valve studies, commonly with the use of stem cells, have not exhibited consistent results. The major issue is that the implanted valve with the seeded cells causes an over-abundance of tissue growth, ultimately leading to structural valve failure due to events such as leaflet fusion with the surrounding vasculature [22]. Recently, several studies have exhibited the importance of elastin within the ECM material to trigger in vivo chemotaxis, leading to native cardiovascular cell migration to the site of the implant [15,16,17,18,19,20]. However, there are still some limitations associated with effective bio-scaffold ECM-to-native cell communications if the scaffold contains xenogeneic elastin. While allogeneic scaffolds have shown promising regenerative findings [21], including for the heart valve application, there is a limitation in being able to harvest bio-scaffolds from human donors.

We were able to regenerate a thin layer of engineered allogeneic (human) ECM on PSIS bio-scaffold tubular valves that was rich in elastin content (Figure 2 and Figure 3). Of relevance to our study, the reason why we had utilized PSIS tubular bio-scaffolds was because PSIS tubular bio-scaffolds (Cormatrix, Roswell, GA, USA), have previously been directly used as a compassionate care-based, heart valve replacement implant in infants suffering from CCVD [3]. Elastin has been reported to not be easily produced in vitro [30], but we have demonstrated it is feasible to produce physiologically close levels of elastin (~8% in content) via physiologically relevant oscillatory flow conditioning (OSI = 0.20 [25]) of hBMMSC-seeded PSIS bio-scaffolds, noting that native heart valves consist of roughly 10% elastin [31]. More specifically, we found that our oscillatory flow conditioned, bio-scaffold tubular valves that had been seeded with hBMMSCs had a significantly higher (*p* < 0.05) ECM composition, which consisted of mucins, elastin, and fibrin but there was no significant difference (*p* > 0.05) between the groups for collagen. Mucins, collagen, and elastin play a critical role in the structure and function of heart valves; furthermore, elastin is a key component in promoting cardiovascular regeneration [15,16,17,18,19,20]. Elastin is crucial in accelerating valve regeneration rates after implantation, since elastin is a known driver of chemotaxis and beneficial tissue remodeling [16]. Our study here clearly demonstrated that physiologically relevant magnitudes of both fluid-induced oscillations and shear stresses [13] allowed for the development of enhanced engineered heart valve matrix with significantly higher (*p* < 0.05) and robust elastin content (Figure 2 and Figure 3). In particular, the augmented elastin in the engineered ECM deposited onto the valve could serve as a trigger for chemotaxis [32] and hence accelerate regeneration, which we previously reported needs to occur within 3 months following implantation in a non-human primate model [11]. 

To prevent the seeded hBMMSCs from causing an adverse outcome in an allogeneic, i.e., human recipient, such as leaflet fusion [22] following valve implantation, the cells would first need to be removed from the engineered ECM. Our decellularization protocol proved to be able to safely remove the hBMMSCs (Figure 4). The decellularized samples had little to no cells present on them. The control group and the decellularized group (Figure 4) were analyzed to quantify collagen and elastin content. It was found that there was no significant difference (*p* > 0.05) in elastin and collagen amounts between the control and decellularized groups (Table 1). Furthermore, there were no significant differences (*p* > 0.05) between the groups with regard to linear stiffness or maximum tensile stress in the specimen length and width directions (Appendix A). This indicates that while the decellularization protocol can remove cells from a bio-scaffold (Figure 4), it does not impact its structure (Table 1) or function (Appendix A). Finally, after being seeded with VICs, the elastin-rich engineered ECM facilitated a significantly higher collagen secretion (*p* < 0.05) by the cells relative to equivalent VIC seeding on raw PSIS bio-scaffolds (Figure 5). This thereby provides preliminary evidence that supports our hypothesis that the elastin-rich engineered ECM can accelerate valve tissue regeneration.

In conclusion, in this study, we produced elastin-rich engineered valve ECM in vitro under dynamic, physiologically relevant oscillatory flow stem cell culture conditions, with stem cells seeded on a bio-scaffold [13,23]. The stem cells were subsequently safely decellularized with the retention of the elastin-rich content within the engineered tissues. This elastin-rich valve ECM was able to enhance the collagen secretion by VICs. This serves as preliminary evidence that the elastin-rich component in these engineered tissues can facilitate accelerated valve regeneration when this study transitions next to an in vivo model for assessing its somatic growth potential for the eventual treatment of CCVD in the young. We recognize that while our study is still preliminary in its findings, we believed that it was important to present these findings now, because producing allogeneic elastin is not straightforward, and it’s a true novelty of our current study. Nonetheless, three major limitations of our study included the fact that: (1) VICs producing in vitro collagen is not directly indicative of how well heart valve tissues will regenerate especially given the important role of the immune system in regulating the valve tissues post-implantation. Therefore, we aim to assess this in vivo in our immediate next step moving forward in a juvenile nonhuman primate animal model. (2) while we accept that our sample sizes in this study are relatively small, we have confirmed the presence of increased amounts of elastin in hBMMSC-produced engineered tissues that was deposited on the bio-scaffold valves, following more than one run of utilizing the 0.2 OSI setting in our torpedo bioreactor for hBMMSC-seeded dynamic culture (Appendix A). (3) While the raw tubular PSIS bio-scaffold valve can withstand systemic pressures [11], it would be important to confirm that our valves consisting of the allogenic elastin-rich engineered tissues deposited onto the tubular PSIS bio-scaffold, with the hBMMSCs decellularized, are able to as well. This is a part of our ongoing efforts to assess the hydrodynamics and durability of our elastin-rich valve constructs, following hBMMSCs decellularization. We are hereby truly motivated to hopefully reach first in human clinical trials in about five years to help infants born with CCVD who will hopefully be able to have a normal, healthy life following a heart valve replacement surgery, with our decellularized elastin-rich valve.

## 5. Patents

Title of Invention: Materials and Methods for Accelerating Cardiovascular Tissue Regeneration. Inventors: Sharan Ramaswamy, Brittany Gonzalez, Ariadna Herrera, Alexander Williams. Status: Issued, Non-Provisional US patent, patent number, 11376347.

## Figures and Tables

**Figure 1 bioengineering-09-00587-f001:**
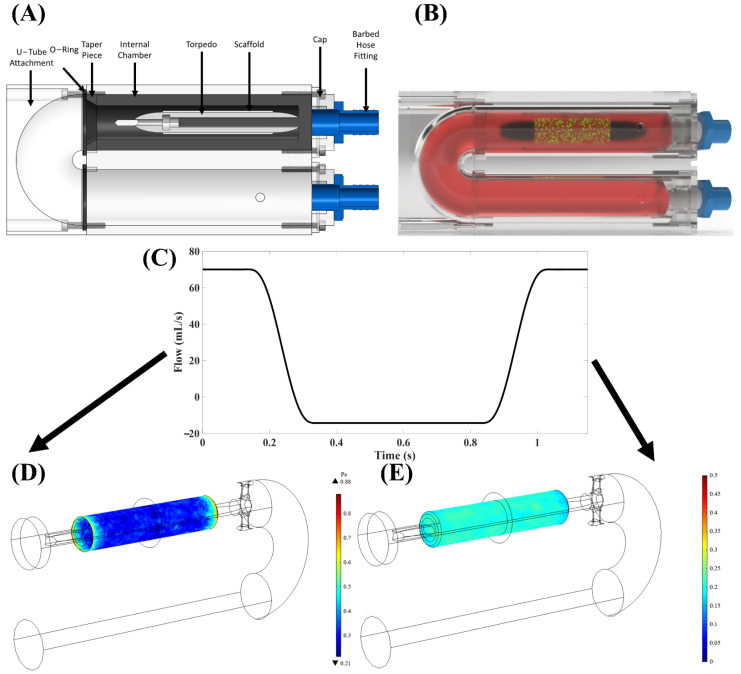
Developed in-house “torpedo” bioreactor with computational fluid dynamics (CFD; ANSYS Workbench, ANSYS Inc., Canonsburg, PA, USA). (**A**) A design of our bioreactor with the major parts, including the torpedo and tube inlet and outlet. (**B**) Bioreactor schematic filled with cell culture media, the torpedo, and a bio-scaffold on the torpedo with seeded human stem cells represented by the green dots. (**C**) A square waveform was used with a 17% forward and 83% backward flow. This waveform provided a specimen (**D**) time-averaged wall shear stress (TAWSS) of 3.1 dynes/cm^2^ and an (**E**) OSI of ~0.20.

**Figure 2 bioengineering-09-00587-f002:**
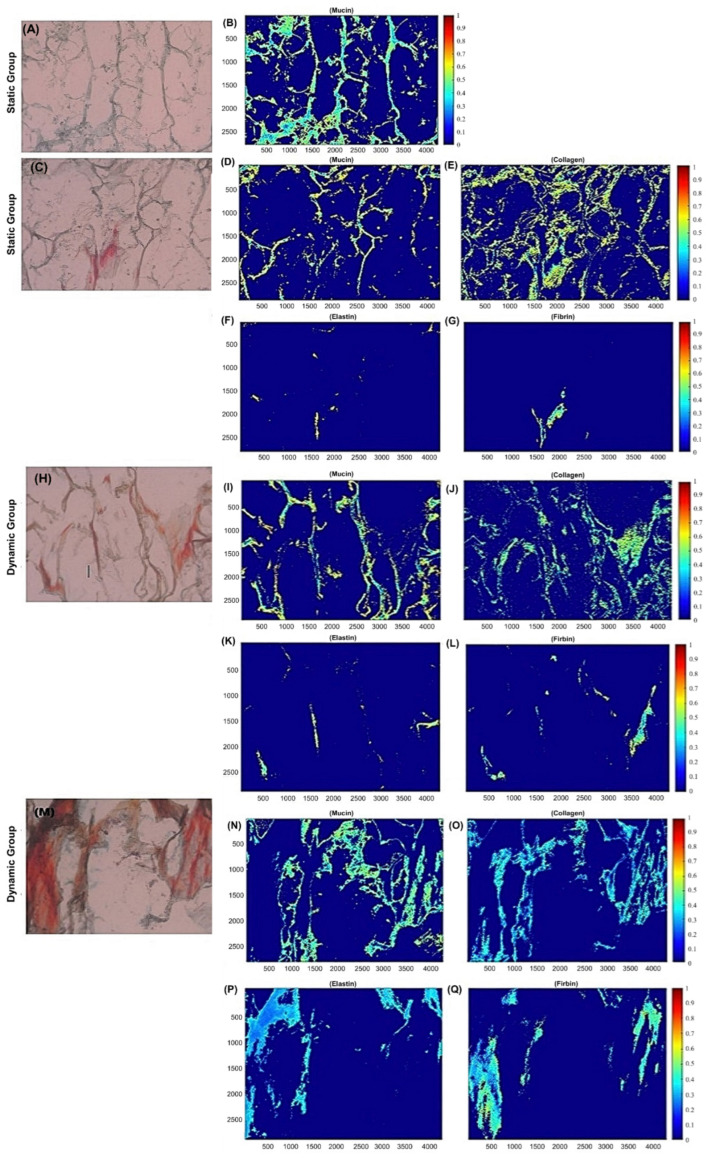
ECM quantification of static and dynamic PSIS valves. (**A**,**C**) Static PSIS valves were assessed for ECM quantification via spatial intensity maps (**B**,**D**–**G**). Spatial intensity maps for these static PSIS valves at a depth of (**A**) 16 µm and (**C**) 304 µm were computed. At the outer surface of the PSIS valve, there was (**B**) 21% mucins present with no other ECM components. At a deeper depth, closer to the inner surface of the PSIS valve, there was a presence of (**D**) 9% mucins, (**E**) 20% collagen, (**F**) 0.6% elastin and (**G**) 1% fibrin. (**H**,**M**) Dynamic PSIS valves were also assessed for ECM quantification via spatial intensity maps (**I**–**L**,**N**–**Q**). Spatial intensity maps for these dynamic PSIS vales at a depth of (**H**) 16 µm and (**M**) 304 µm were computed. At the outer surface of the PSIS valve, there was (**I**) 14% mucins, (**J**) 16% collagen, (**K**) 1% elastin and (**L**) 4% fibrin present. At a larger depth, closer to the inner surface of the PSIS valve, there was a presence of (**N**) 21% mucins, (**O**) 16% collagen, (**P**) 9% elastin and (**Q**) 11% fibrin. Magnification 50×.

**Figure 3 bioengineering-09-00587-f003:**
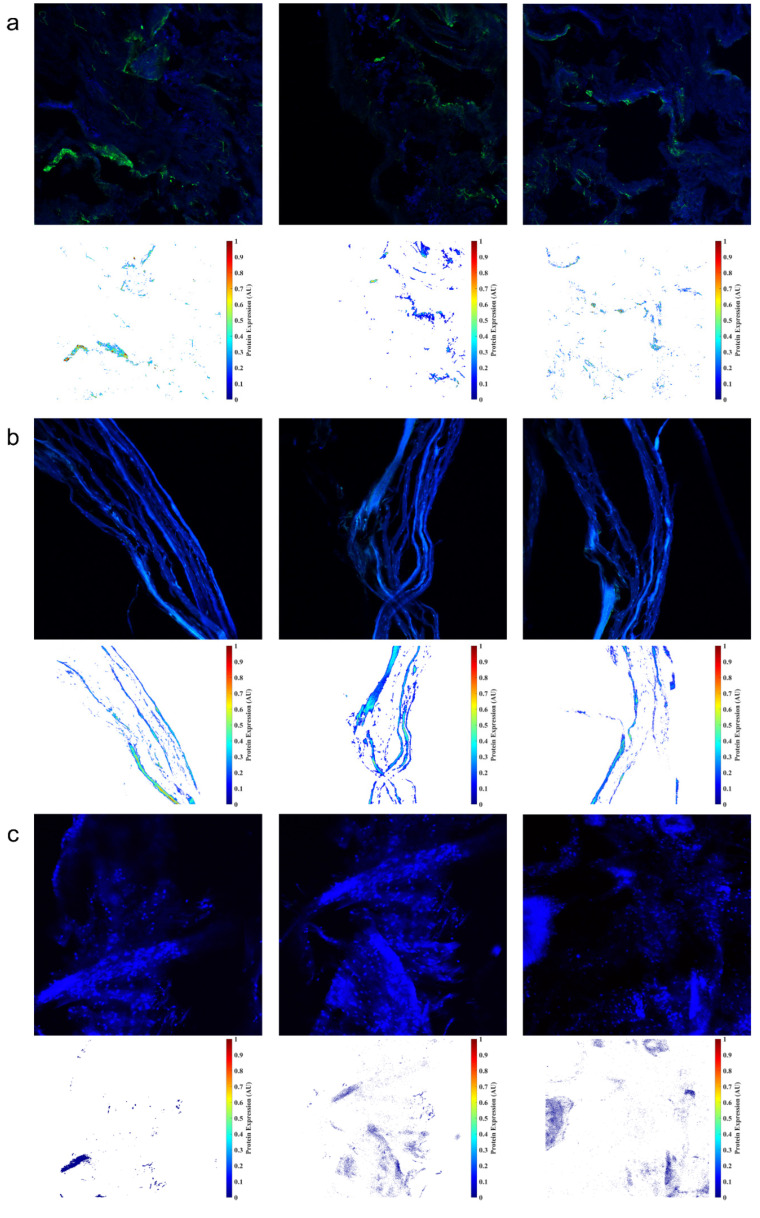
Elastin immunofluorescent imaging and spatial intensity mapping for: (**a**) seeding hBMMSCs onto PSIS bio-scaffold substrates, rotisserie cultured for 8 days followed by an additional 14 days of oscillatory flow conditioning (TAWSS of 3.1 dynes/cm^2^ and an OSI of 0.20), (**b**) seeding hBMMSCs onto PSIS bio-scaffold substrates and then statically (rotisserie) cultured for 22 days, and (**c**) the raw PSIS bio-scaffold.

**Figure 4 bioengineering-09-00587-f004:**
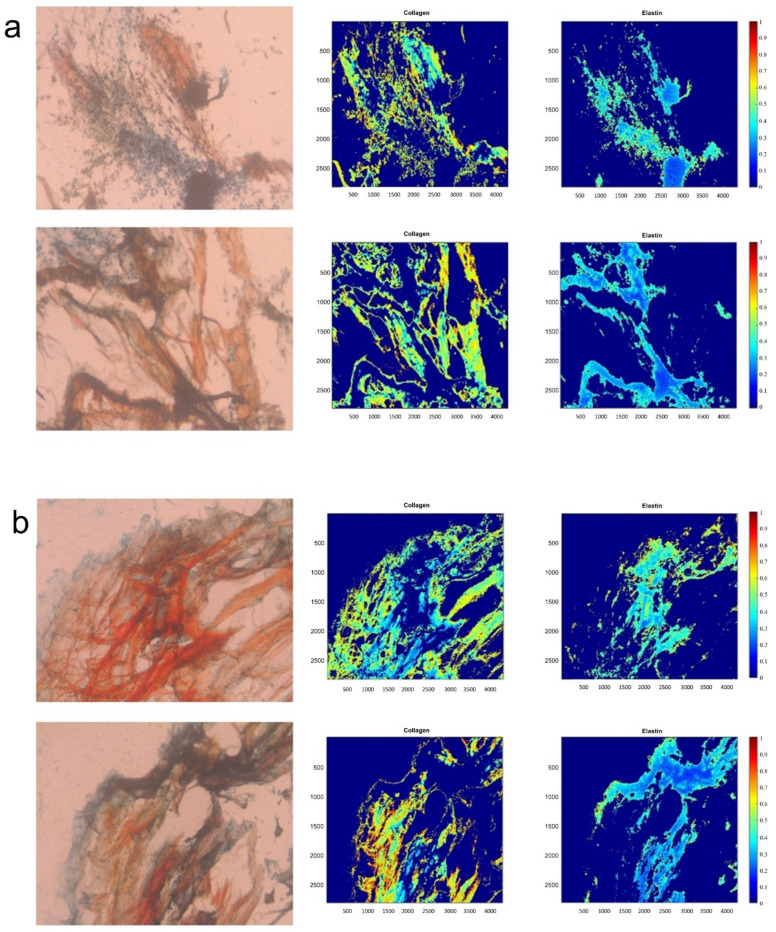
ECM quantification of the control (untreated) and decellularized elastin-rich engineered valve tissues. (**a**) Images of the Movat’s stained control group and (**b**) of the Movat’s stained decellularized group were assessed for quantification of ECM collagen and elastin spatial intensity maps.

**Figure 5 bioengineering-09-00587-f005:**
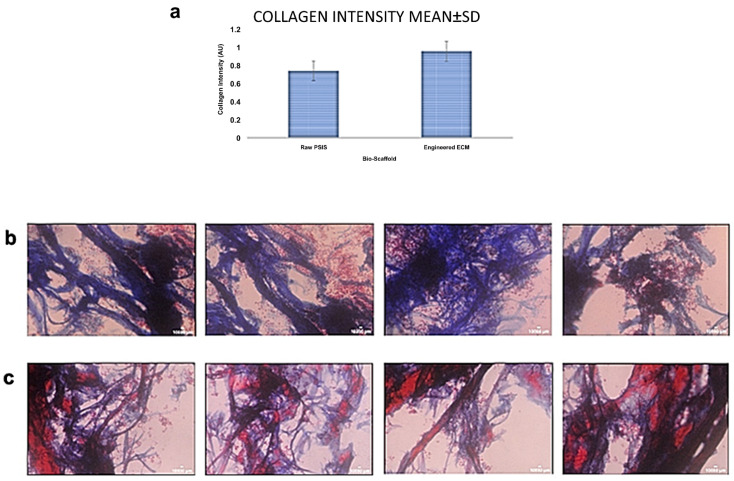
Rat VICs collagen secretion quantification and Masson’s Trichrome staining after seeding and subsequent culture for 8 days of static (rotisserie) culture and 7 days of oscillatory flow conditioning (TAWSS of 3.1 dynes/cm^2^ and an OSI of 0.20; *n* = 4 specimens/group). (**a**) Average ± SD of collagen secretion by rat VICs seeded on the elastin-rich engineered ECM versus raw PSIS bio-scaffolds. Masson’s Trichrome histological images of the *de novo* collagen secretion by rat VICs on the outer substrate surface (sectioned at 16 μm from the outer surface) where the cells were seeded (*n* = 4/group) on the: (**b**) engineered elastin-rich engineered ECM and (**c**) raw PSIS bio-scaffolds.

**Table 1 bioengineering-09-00587-t001:** Average % ± SEM of elastin and collagen on control and decellularized scaffolds. It was found that there was no significant difference (*p* > 0.05) in elastin and collagen amounts between the two groups.

Scaffold Group	Elastin (%)	Collagen (%)
Control	19.60% ± 3.39	26.40% ± 2.09
Triton X-100 Treated	22.06% ± 1.95	31.29% ± 2.73
*p*-value	0.57 (>0.05)	0.18 (>0.05)

## Data Availability

Kindly send a request via e-mail to the corresponding author, Sharan Ramaswamy at: sramaswa@fiu.edu.

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
