# Peer review of "Stem Cell-Secreted Allogeneic Elastin-Rich Matrix with Subsequent Decellularization for the Treatment of Critical Valve Diseases in the Young"

_bioengineering, 2022, doi:10.3390/bioengineering9100587_

Round 1

Reviewer 1 Report

Very interesting findings. I have a few comments and queries:

1.Labelling incorrect in Fig. 2, letter in figure does not correspond to the letter (substance mentioned) in the legend, please correct.

2.Is elastin integrated in the valve scaffold or is it as a layer on top? please clarify (maybee Fig S5 has the answer, should than be in the paper instead of supplement).

3.The PSIS in Fig S2 is all tissue valve tissue, or is this a vessel with valve tissue like an aortic root, please clarify.

4.Are the histology specimens taken from the valve itself or are they from the vessel wall? 

5.Will the scaffold disappear with time, absorbable?, and how long will that take in humans?

6.In which way will the scaffold not induce inflammation if it is not absorbed?

7.Are the findings applicable only to atrioventricular valves? 

8.Will the new ECM withstand systemic pressure?

Author Response

Kindly refer to the uploaded file.  Thank You.

Reviewer 2 Report

In general it is a well structured and well written article. The content is quite interesting and the introduction in very helpful even for non experts in the field. The Materials and methods introduces the reader to the way the researchers approached the flow profile and the significance of oscillatory shear index. The seeding in the two groups of valves (static & dynamic) is adequately presented as well as their histology assessment. The sample in both groups is quite small, however the statistical analysis is fair. The results are quite analytic with the great support of the images. The discussion points out the most important findings and is well referenced. The conclusion is in relevance to the findings.

The authors also clearly state the major limitations of the study.

Author Response

(The authors gave the same response as above.)

Reviewer 3 Report

No additional suggestions.

Well done, merit publication in the current version.

Author Response

(The authors gave the same response as above.)
